# Polymorphisms within the SARS-CoV-2 Human Receptor Genes Associate with Variable Disease Outcomes across Ethnicities

**DOI:** 10.3390/genes14091798

**Published:** 2023-09-14

**Authors:** Theolan Adimulam, Thilona Arumugam, Anushka Naidoo, Kogieleum Naidoo, Veron Ramsuran

**Affiliations:** 1School of Laboratory Medicine and Medical Sciences, College of Health Sciences, University of KwaZulu-Natal, Durban 4041, South Africa; theoadimulam@gmail.com (T.A.); cyborglona@gmail.com (T.A.); 2Centre for the AIDS Programme of Research in South Africa (CAPRISA), Nelson R. Mandela School of Medicine, University of KwaZulu-Natal, Durban 4041, South Africa; anushka.naidoo@caprisa.org (A.N.); kogie.naidoo@caprisa.org (K.N.); 3South African Medical Research Council (SAMRC), Durban 4013, South Africa

**Keywords:** COVID-19, SARS-CoV-2 viral load, ACE2, TMPRSS2, NRP1, CD147, SNP, disease severity

## Abstract

The contribution of human genes to the variability of disease outcomes has been shown to be important across infectious diseases. Studies have shown mutations within specific human genes are associated with variable COVID-19 outcomes. We focused on the SARS-CoV-2 receptors/co-receptors to identify the role of specific polymorphisms within *ACE2*, *TMPRSS2*, *NRP1* and *CD147*. Polymorphisms within *ACE2* (rs2285666), *TMPRSS2* (rs12329760), *CD147* (rs8259) and *NRP1* (rs10080) have been shown to associate with COVID-19 severity. Using cryopreserved samples from COVID-19-positive African, European and South Asian individuals within South Africa, we determined genotype frequencies. The genetic variant rs2285666 was associated with COVID-19 severity with an ethnic bias. African individuals with a CC genotype demonstrate more severe COVID-19 outcomes (OR = 7.5; 95% CI 1.164–80.89; *p* = 0.024) compared with those with a TT genotype. The expressions of *ACE2* and SARS-CoV-2 viral load were measured using droplet digital PCR. Our results demonstrate rs2285666 and rs10080 were significantly associated with increased SARS-CoV-2 viral load and worse outcomes in certain ethnicities. This study demonstrates two important findings. Firstly, SARS-CoV-2 viral load is significantly lower in Africans compared with individuals of European and South Asian descent (*p* = 0.0002 and *p* < 0.0001). Secondly, SARS-CoV-2 viral load associates with specific SARS-CoV-2 receptor variants. A limited number of studies have examined the receptor/co-receptor genes within Africa. This study investigated genetic variants within the SARS-CoV-2 receptor/co-receptor genes and their association with COVID-19 severity and SARS-CoV-2 viral load across different ethnicities. We provide a genetic basis for differences in COVID-19 severity across ethnic groups in South Africa, further highlighting the importance of further investigation to determine potential therapeutic targets and to guide vaccination strategies that may prioritize specific genotypes.

## 1. Importance

This study addresses an important knowledge gap in the genetic understanding of COVID-19 severity in South Africa. We show that the frequency of genetic mutations within viral receptor genes is variable across different ethnic groups and may influence COVID-19 severity. More importantly, we show that SARS-CoV-2 viral load varies on the basis of genetic variants across ethnic groups in South Africa. This study highlights the importance for further investigation of the role of genetics in determining potential therapeutic targets and to guide COVID-19 vaccination strategies and the allocation of limited resources.

## 2. Introduction

Coronavirus disease 19 (COVID-19) is caused by the severe acute respiratory syndrome coronavirus-2 (SARS-CoV-2). Early in the COVID-19 pandemic, there was considerable concern over regions with a high burden of infectious diseases, such as Africa [1,2]. Furthermore, it was predicted that African regions would be impacted far more than European and American regions due to weaker healthcare infrastructure and socioeconomic factors [3]. As of 19 April 2023, the World Health Organization (WHO) has reported 7,584,874 confirmed cases and 178,597 deaths due to COVID-19 in Africa. The impact of COVID-19 in Africa has been relatively lower compared with Europe, the Americas and Asia, based on confirmed cases and excess deaths reported [4,5,6]. COVID-19 severity has been variable across populations globally. It has been shown that key factors, such as age, gender and underlying comorbidities are associated with COVID-19 severity and outcomes [7].

There is growing evidence that genetic factors also contribute to the severity of COVID-19 [8,9]. Disease severity has been categorized according to the symptoms experienced by COVID-19-positive individuals. A majority of individuals experience mild or moderate symptoms, while some advance to severe disease requiring clinical intervention and ultimately succumb to the disease [10]. It has been shown that differences in immunity, gene expression and genetic diversity contribute to differences in susceptibility and severity of COVID-19 [8]. Beyond this, genetic studies on infectious diseases, such as SARS-CoV-2, are important for understanding the dynamics of disease transmission and identifying risk factors that confer greater susceptibility to severe disease and worse outcomes. Furthermore, these studies may guide the development of vaccines, drug development strategies and public health planning for the prevention and control of future disease outbreaks [8,11,12].

Several studies have examined host genetic effects on COVID-19 disease susceptibility and severity, thus identifying varying effects on disease severity between populations globally [13,14,15,16]. The genetic factors associated with COVID-19 outcomes have been widely studied within the American, European and Asian continents. However, genetic studies to understand COVID-19 severity within the African continent have been limited [17,18]. This has contributed to the gap in understanding the impact of genetic diversity and unique genomic makeup of African populations on COVID-19, as well as many other infectious diseases that disproportionately affect African individuals [19,20]. Therefore, this study aims at investigating the effect of single-nucleotide polymorphisms (SNPs) within select SARS-CoV-2 human receptor/co-receptor genes and their association with COVID-19 severity within an African cohort.

SARS-CoV-2 infection and replication requires entry into human cells, primarily those within the respiratory system [21]. The SARS-CoV-2 spike protein subunit 1 (S1) interacts with the host receptors angiotensin-converting enzyme 2 (ACE2) to gain entry into cells [22,23,24,25]. This interaction is dependent on the proteolytic cleavage of the spike protein by transmembrane serine protease 2 (TMPRSS2) to release the S1 and S2 subunits [21,26,27]. In addition to these receptors, neuropilin-1 (NRP1) and Basigin (CD147) have been shown to facilitate the entry of SARS-CoV-2 independently of ACE2 and TMPRSS2 [28,29]. Polymorphisms within these genes may influence the ability of SARS-CoV-2 to enter the human cell and be determinants for the disease severity of COVID-19 [30]. This phenomenon has been shown previously with regard to HIV infection. Samson et al. (1996) showed that a mutation within the HIV receptor chemokine receptor 5 (CCR5) conferred protection against HIV infection [31]. The mutant allele within CCR5 results in the deletion of 32 nucleotides (CCR5-∆32), which results in the production of a truncated protein, hampering viral entry [32,33].

With regard to SARS-CoV-2, SNPs within the SARS-CoV-2 host receptor genes have demonstrated detrimental and protective effects across ethnic groups [34]. Within the *ACE2* and *TMPRSS2* genes, rs2285666 and rs12329760, respectively, have been identified to be associated with COVID-19 severity in some studies, while others have reported no association [35,36]. These SNPs were chosen based on data from an Egyptian cohort which showed that rs2285666 (*ACE2*) and rs12329760 (*TMPRSS2*) were significantly associated with the severity of COVID-19 and select co-morbidities [36].

With regard to *CD147* and *NRP1*, the rs8259 and rs10080 SNPs were chosen because they are found within the 3′-UTR of their respective genes. Furthermore, these SNPs are found within predicted micro-RNA binding sites, with rs8259 (*CD147*) being a predicted binding point for miR-492 and rs10080 (*NRP1*) being a predicted binding point for miR-338 [30,37,38,39]. We recently reviewed 42 SNPs within *ACE2* (12), *TMPRSS2* (10), *NRP1* (15) and *CD147* (5) and found that rs2285666 (ACE2), rs12329769 (TMPRSS2), rs10080 (NRP1) and rs8259 (CD147) are significantly associated with COVID-19 co-morbidities within European and South Asian settings [9]. We selected these SNPs as a starting point to investigate their association with COVID-19 severity and susceptibility in an African setting.

In this study, we investigated the effect of SNPs within SARS-CoV-2 receptors and co-receptors on COVID-19 severity; furthermore, we compared the SARS-CoV-2 viral load across ethnic groups within South Africa.

## 3. Methods

### 3.1. Study Design

A cohort of SARS-CoV-2 polymerase chain reaction (PCR)-positive individuals were considered to participate in this study. SARS-CoV-2 positivity was determined using the TaqPath COVID-19 RT-PCR kit (Thermo Fisher Scientific, Waltham, MA, USA; cat no. A48102) and the QuantStudio 5 Real-Time PCR system (Applied Biosystems, Woburn, MA, USA), as per the manufacturer’s instructions.

A longitudinal cohort (SARS-CoV-2 Antibody Prevalence Study (SAP; *n* = 591)) of SARS-CoV-2-positive individuals were successfully recruited into this study, after signing the informed consent form. Only individuals with an RT-qPCR SARS-CoV-2-positive result were recruited. Buffy coat samples (*n* = 591) were collected 6 weeks, 3 months and 6 months post infection, between 2021 and 2022. However, only samples from the first time point (*n* = 560; 6 weeks post infection) were used in this study. All demographic and clinical data were collected after the study protocol, and consent forms were signed by each participant. All participant details and samples were collected with unique patient identifiers. The ethnic breakdown for this study was South African Black (*n* = 290), Indian (*n* = 246) and Caucasian (*n* = 24) individuals within South Africa. We scored each patient into two disease states (Table 1): no clinical presentations and those with clinical presentations, based on the presence of specific clinical presentations (including asymptomatic, symptomatic and oxygen required). Furthermore, we grouped ethnicity according to each of the two disease states (Table 2).

In addition to the buffy coat samples, SARS-CoV-2-positive nasopharyngeal swabs (*n* = 117) were obtained for an unbiased fraction of the total number of participants within this study.

Ethical approval for this study was obtained from the Biomedical Research Ethics Committee (BREC) at the University of KwaZulu-Natal, protocol reference number: BREC/00002648/2021.

### 3.2. SNPs Included in the Study

We identified rs2285666 (*ACE2*), rs12329760 (*TMPRSS2*), rs10080 (*NRP1*) and rs8259 (*CD147*). Table 3 shows the minor allele frequency (MAF) of the four SNPs within African, South Asian and European individuals included in this study. Using the Sample size Calculator, we determined the number of samples required [40]. Briefly, using the MAF frequency of the SNPs (Table 3), we determined the sample size for rs2285666 (ACE2 (*n* = 88)), rs12329760 (TMPRSS2), rs10080 (NRP1 (*n* = 62)) and rs8259 (CD147 (*n* = 146)), with α at 0.05 and the power determined at 80%. The calculated sample sizes are all below the number of samples recruited into the study per ethnic group (Table 2).

To investigate genetic variation within *ACE2*, *TMPRSS2*, *NRP1* and *CD147*, DNA was extracted from 200 µL of buffy coat samples using the Quick-DNA Miniprep Plus Kit (Zymo, Irvine, CA, USA) as per the manufacturer’s instructions; the extracted DNA was standardized to 50 ng/µL and stored at −20 °C. The allelic examination of rs2285666, rs12329760, rs10080 and rs8259 was carried out using a Real-Time PCR (RT-PCR) protocol and TaqMan genotyping probes (Thermo Fisher Scientific, Waltham, MA, USA) on the QuantStudio 5 instrument as per manufacturer’s guidelines. Briefly, TaqMan Genotyping Master Mix (ThermoFisher Scientific) was prepared as a 5 µL reaction as per manufacturer guidelines, and RT-PCR was performed. RT-PCR cycling conditions and catalogue numbers for rs2285666 (C > T), rs12329760 (C > T), rs10080 (A > G) and rs8259 (T > A) are available on request.

### 3.3. ACE2, NRP1 and SARS-CoV-2 Viral Load Quantification

Nasopharyngeal swabs were collected and stored at 4 °C. RNA was extracted using the AmoyDx Virus/Cell RNA kit (Amoy Diagnostics, Xiamen, China) and then standardized to 20 ng/µL. Thereafter, cDNA was prepared using the Superscript VILO cDNA synthesis kit (ThermoFisher Scientific) according to manufacturer guidelines and then stored at −20 °C.

SARS-CoV-2 viral load was determined using the TaqPath COVID-19 RT-PCR kit and the QIAcuity Probe PCR kit (Qiagen, Hilden, Germany) on the QIAcuity Digital PCR instrument (QIAcuity Nanoplate 26K 24-well plate), as per the manufacturer’s specifications. Briefly, QIAcuity Probe master mix was prepared with the TaqPath COVID-19 RT-PCR kit probe and a 10 µL of sample cDNA for a total of 40 µL. SARS-CoV-2 viral load was determined and reported as copies/µL.

Gene expression of *ACE2* and *NRP1* was measured using QX200 ddPCR EvaGreen Supermix (Bio-Rad Laboratories, Hercules, CA, USA) on the QX200 droplet digital PCR (ddPCR) instrument, as per manufacturer’s instructions. Briefly, cDNA from nasopharyngeal swabs was added to EvaGreen master mix (total 40 µL) that was prepared according to manufacturer guidelines. Droplet generation was performed using the DG32 cartridge and gasket (Bio-Rad) set with 70 µL of droplet generation oil for EvaGreen (Bio-Rad). Primer sequences are available on request.

### 3.4. Statistical and Bioinformatics Analysis

GraphPad Prism 8 software was used for analysis. Mann–Whitney U test was used to compare continuous variables; categorical variables were analyzed using Fisher’s exact tests. A *p*-value of less than 0.05 was considered statistically significant.

## 4. Results

We found that a large proportion of African individuals present with no clinical presentations compared with European and South Asian individuals. In addition, South Asians account for the largest proportion of individuals with clinical presentations (Table 2).

### 4.1. Variants within SARS-CoV-2 Receptor Genes Are Associated with COVID-19 Severity across Ethnic Groups

We investigated the effect of SNPs within the SARS-CoV-2 receptor genes on COVID-19 severity in PCR-confirmed-positive COVID-19 individuals. The genotype frequency of each SNP was determined for each ethnic group across disease states (Appendix A). Owing to the lower sample number of European individuals (*n* = 24), we excluded European individuals from further genotype analysis.

We found different associations between individuals with no clinical and clinical presentations of COVID-19 across African and South Asian individuals for each polymorphism. The counts for genotype, odds ratio (OR), 95% confidence interval (CI) and *p*-values are summarized in Figure 1 below (Appendix A). For African individuals, the CC genotype compared with the TT genotype of rs2285666 is significantly detrimental towards clinical presentations of COVID-19 (OR = 7.5; 95% CI 1.164–80.89; *p* = 0.024, Figure 1A). However, this variant does not show significance within South Asians. Interestingly, we found the CC genotype of rs12329760 was significantly protective in African individuals (OR = 0.3134; 95% CI 0.1222–0.8045; *p* = 0.024 Figure 1A) but not in South Asian individuals.

### 4.2. SARS-CoV-2 Viral Load Is Different across Ethnic Groups

The variability of SARS-CoV-2 viral load between different ethnic groups has been shown previously [41]. This study examines SARS-CoV-2 viral loads in African, South Asian and European individuals within South Africa. African (mean Log VL = −0.4683) individuals have significantly lower mean SARS-CoV-2 viral loads compared with European (mean Log VL = 1.421) and South Asian (mean Log VL = −2.145) individuals, respectively (*p* = 0.0002 and *p* < 0.0001; Figure 2). However, we did not find a significant difference in SARS-CoV-2 viral load between South Asian and European individuals (*p* = 0.4010; Figure 2).

### 4.3. SARS-CoV-2 Viral Load Associates with rs2285666 (ACE2) and rs12329760 (TMPRSS2) in African Individuals

We next examined the effect of variants within the SARS-CoV-2 receptors on viral load in African individuals. To our knowledge, this is the first study to show the association of the ACE2 SNP rs2285666 and the TMPRSS2 SNP rs12329760 with SARS-CoV-2 viral load in African individuals. The number of patients within the South Asian group was too low to conduct any statistical analysis (Appendix A). The CC genotype of rs2285666 is associated with significantly higher viral load compared with the CT + TT genotypes (*p* = 0.0136; Figure 3A). In addition, the CC genotype of rs12329760 is significantly associated with lower viral load compared with the CT + TT genotypes (*p* = 0.0361; Figure 3B). On the contrary, we did not find an association between rs10080 and rs8259 with SARS-CoV-2 viral load (*p* = 0.0725 and *p* = 0.6417; Figure 3C,D).

### 4.4. ACE2 and NRP1 Expression and SARS-CoV-2 Viral Load in African Individuals

Due to sample limitations, the measurement of mRNA expression was prioritized to ACE2 and NRP1, since these genes showed significance and they serve as receptors to the virus and not co-receptors. ACE2 is known as the key mediator of SARS-CoV-2 infection. Studies have shown that the binding affinity of spike glycoprotein to ACE2 is 20-fold higher than that of its predecessor, SARS-CoV [42]. Furthermore, due to the association of rs2285666 and SARS-CoV-2 viral load in African individuals, we investigated the effect of *ACE2* expression on SARS-CoV-2 viral load in nasopharyngeal swabs at the time of infection. Interestingly, we found that *ACE2* negatively correlates with SARS-CoV-2 viral load (r = −0.3488; *p* = 0.0020; Figure 4A) across ethnic groups. In addition, we found no correlation between ACE2 copies and rs2285666 (Figure 4B). Lastly, we found no significant correlation between *NRP1* expression and SARS-CoV-2 viral load (r = −0.0587; *p* = 0.6529; Figure 4C). In addition, *NRP1* expression trends with rs10080 genotypes across South African individuals; however, these results were not significant (AA vs. GG; *p* = 0.3691; Figure 4D).

## 5. Discussion

This study aimed to investigate the effect of SNPs with SARS-CoV-2 receptors and co-receptors on COVID-19 severity; furthermore, we compared SARS-CoV-2 viral load across ethnic groups within Durban, South Africa. To our knowledge, this is the first study to evaluate SARS-CoV-2 receptor variants and viral load within a South African setting. We observed that specific variants (rs2285666 and rs10080) are significantly associated with increased SARS-CoV-2 viral load and worse outcomes in certain ethnicities. This study identified two significant findings. Firstly, we identified that Africans have significantly lower SARS-CoV-2 viral loads compared with European and South Asian individuals. Secondly, SARS-CoV-2 viral load correlates with specific SARS-CoV-2 receptor variants in African individuals.

Previous studies have highlighted differences in SARS-CoV-2 viral loads across different ethnic groups. Peterson et al. (2021) showed that average viral loads (lower cyclic thresholds/numbers) were lower in African Americans compared with Caucasian American and Asian individuals [41]. In line with previous reports, we show that SARS-CoV-2 viral load is significantly lower in Africans compared with European and South Asian individuals, respectively. The difference in viral load between these ethnic groups may be explained by genetic variants with the SARS-CoV-2 host receptor genes (*ACE2*, *TMPRSS2*, *NRP1* and *CD147*).

Several studies have investigated the association of rs2285666 (ACE2) and rs12329760 (TMPRSS2) with COVID-19 severity [35,36,43,44,45]. Interestingly, these studies have either confirmed [13,36,46,47,48,49,50,51] or contradicted [52,53,54,55] the association between rs2285666 and/or rs12329760 with COVID-19 severity. These differing observations have been attributed to insufficient sample sizes and variation in study designs; however, ethnic variation between cohorts and study populations can also explain these conflicting results [56].

In agreement with previous data, we found that the CC genotype of rs2285666 is significantly associated with clinical presentations of COVID-19 among individuals of African descent. Consistent with these results, we show that the CC genotype of rs2285666 is significantly associated with higher SARS-CoV-2 viral load compared with the CT + TT genotypes in individuals of African descent (*p* = 0.0136), further supporting the relationship between rs2285666 (CC) and severe disease. In contrast, we found no significant association between rs2285666 (CC vs. TT) and clinical presentations of COVID-19 severity for individuals of South Asian descent.

Several studies have investigated the TMPRSS2 variant (rs12329760) and COVID-19 severity across different Asian and European groups [13,46,47,48,50,54,57,58,59,60,61]. In this study, we found a significant association between the CC genotype of rs12327960 and no clinical presentations of COVID-19 in African individuals. Furthermore, we showed that the CC genotype is significantly correlated with lower SARS-CoV-2 viral loads in Africans. These findings are consistent with studies that confirm the association between rs12329760 and COVID-19 severity [36,52]. Despite the frequency of the mutant and wild-type alleles being similar across African, European and South Asian individuals, we found no association between rs12327960 and COVID-19 in South Asian individuals. We show that the CC genotype of rs12329760 has no association with COVID-19 clinical presentations in South Asian individuals (Appendix A). We attribute these observations to the small number of South Asian infected individuals with no clinical presentations (*n* = 26) within this study compared with the large majority who had clinical presentations of COVID-19 (*n* = 220).

Variants within *ACE2* and *TMPRSS2* have been extensively studied regarding COVID-19 severity and susceptibility. Despite subsequent studies identifying *NRP1* and *CD147* as additional host cell receptors for SARS-CoV-2 entry [28,29], few studies have investigated the effect of variants within these genes and COVID-19 severity [29,62,63]. As such, this study is the first to investigate the relationship between variants within the *NRP1* and *CD147* genes and COVID-19 severity. Our data show that the AA genotype of rs10080, a 3′-UTR variant within *NRP1*, is associated with clinical presentations of COVID-19 in African individuals (Appendix A) compared with the AG + GG genotypes. In addition, we show that the A allele is significantly associated with clinical presentations of COVID-19 compared with the G allele (*p* = 0.001). Furthermore, we show that SARS-CoV-2 viral loads are lower for African individuals with the AA genotype than the AG + GG genotypes. Our investigation of the rs8259 (*CD147*) showed no association with COVID-19 severity and viral load across ethnic groups. This finding is consistent with recent meta-analysis data [64].

We also investigated the difference in gene expression of *ACE2* and *NRP1* among South African individuals. Our data showed no significant difference in gene expression of ACE2 and NRP1 across African, European and South Asian individuals (Appendix A). We further investigated the relationship between SARS-CoV-2 viral load and *ACE2* and *NRP1* expression using ddPCR (as copies/μL). We found no significant difference in *ACE2* and *NRP1* expression across African, European and South Asian individuals (Appendix A). We showed that *NRP1* expression is lower in individuals that are homozygous GG compared with homozygous AA (Appendix A). In addition, *NRP1* expression is negatively correlated with SARS-CoV-2 viral load. It is known that rs10080 is within the 3′-UTR of *NRP1*, and it has been reported that the G allele facilitates the binding of miR-338, thus downregulating the expression of *NRP1* [30,37,38]. Further studies are required to validate the epigenetic effect of miR-338 on rs10080 regulation of *NRP1* expression.

Previous studies report a significant positive correlation between transmembrane *ACE2* expression and SARS-CoV-2 viral load in nasopharyngeal swabs [65,66]. However, Nikiforuk et al. showed that (2021) nasopharyngeal expression of ACE2 plays a dual role in SARS-CoV-2-infected individuals [65]. The study showed that the transmembrane isoform of ACE2 is positively correlated, while the soluble isoforms of ACE2 are negatively correlated with viral RNA load after adjusting for age, biological sex and TMPRSS2 mRNA. In line with these findings, we observed a significant negative correlation between SARS-CoV-2 viral loads and soluble ACE2 expression in nasopharyngeal swabs. In addition, Gutierrez-Chamorro et al. (2021) showed that ACE2 enzymatic activity and gene expression decreased over time. Furthermore, Fajnzylber et al. (2020) and To et al. (2020) showed that SARS-CoV-2 viral loads decreased post-symptom onset across different tissues [67,68]. Within South Africa, the delay in testing for COVID-19 may influence our results.

These data are not unexpected due to the use of human recombinant soluble ACE2 (hrsACE2) for the treatment of COVID-19. A recent case report showed that hrsACE2 was able to reduce SARS-CoV-2 load by a factor of 1000–5000 in in vitro experiments [69]. It is well established that *ACE2* is a target of regulated intramembrane proteolysis (RIP) by ADAM17, thus releasing a soluble ACE2 C-terminal [70,71].

The difference in viral load between these ethnic groups cannot be explained by the specific genetic variants examined here with *ACE2*, *TMPRSRS2*, *NRP1* and *CD147* genes alone. We suggest that differences in immunity, genetics and epigenetics contribute to the observed differences. For instance, the delta-32 mutation within *CCR5* confers protection against HIV infection within Caucasian individuals because the mutation is found at a much higher frequency than in Africans and South Asians. In addition, we have shown that DNA methylation of *BST-2* affects *BST-2* expression. The study showed that BST-2 levels inhibit the production of HIV-1 by hindering the release of viral progeny [72]. Therefore, this study provides an understanding of polymorphisms within the SARS-CoV-2 receptors and co-receptors within a South African cohort.

A core limitation of this study is the low sample size. In addition, the distribution of age within the cohort is skewed. This study only considers individuals older than 18 years of age; furthermore, a large number of individuals within this cohort are middle-aged. Moreover, we note that host gene expression does not necessarily reflect post-transcriptional regulation. However, it has been shown that ACE2 transcript expression does reflect protein expression in upper airway tissues [73].

For future studies, we suggest more collaborative work within Africa to fully understand the genetic contribution to SARS-CoV-2 disease severity. This study shows that South Africa provides a diverse setting for host genetic studies. In addition, we support the use of genetic variant testing as a model for infectious disease prioritization. Bubar et al. (2021) suggested the use of a strategy using age and serostatus to prioritize vaccine roll-out and prioritization [74]. We suggest a point-of-care strategy using genotypic data to prioritize vaccination for individuals with genetic variants which are associated with increased susceptibility and severity of disease. This strategy will reduce the burden on the availability of vaccines within resource-limited areas. This strategy has been recently suggested by Bruce and Johnson (2022), who showed that individuals with genetic risk factors should be prioritized for vaccination drives [75].

## 6. Conclusions

Despite early predictions that the COVID-19 pandemic would affect African countries far worse than other well-resourced countries due to the lack of healthcare infrastructure and the high level of poverty, Africa was not as severely affected by the pandemic compared with other European and Asian settings based on reported data. In this study, we show that SARS-CoV-2 viral loads are significantly different across ethnic groups in South Africa. We found that Black African individuals had significantly lower viral loads compared with European and South Asian individuals. In addition, we showed that SNPs within ACE2, TMPRSS2, NRP1 and CD147 have variable associations with COVID-19 severity across these ethnic groups. This study provides strong evidence that the severity of COVID-19 is dependent on host genetic factors.

## Figures and Tables

**Figure 1 genes-14-01798-f001:**
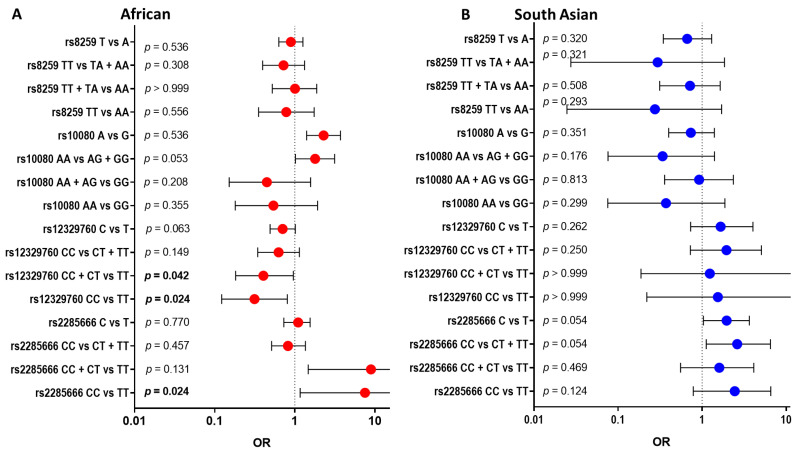
Comparing COVID-19 disease states (no clinical presentations versus clinical presentations) within each of the four SNPs across African and South Asian individuals: The four SNPs included rs2285666 (ACE2), rs12329760 (TMPRSS2), rs10080 (NRP1) and rs8259 (CD147). An odds ratio (OR) of 1 is considered to have no effect (dotted line). Horizontal bars represent the 95% confidence intervals. Alleles or genotypes were used in the Fisher exact analysis, with significance accepted as *p* < 0.05. African individuals (**A**), red dots, showed rs2285666 and rs10080 as significant symptomatic effects. South Asians (**B**), blue dots, showed rs2285666 and rs12329760 as significant symptomatic effects.

**Figure 2 genes-14-01798-f002:**
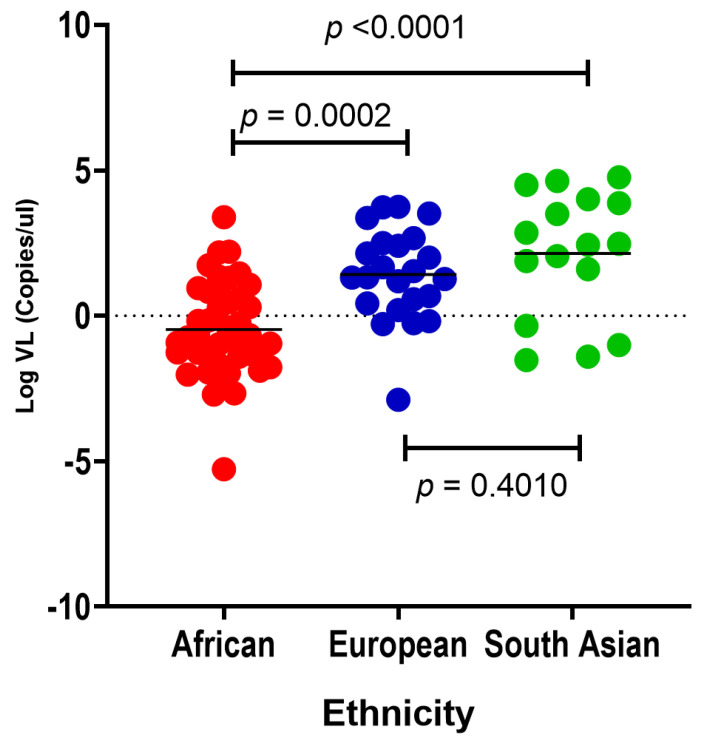
A comparison of nasopharyngeal SARS-CoV-2 viral loads (copies/μL) in African (red dots, *n* = 43), European (blue dots, *n* = 23) and South Asian (green dots, *n* = 16) individuals. Turkey’s multiple comparisons test *p*-values show significant differences between the mean viral loads between African, European and South Asian individuals. The horizontal bars show the *p*-values for the ethnic groups being compared. African individuals have significantly lower SARS-CoV-2 viral load compared with European (*p* = 0.0002) and South Asian (*p* < 0.0001) individuals.

**Figure 3 genes-14-01798-f003:**
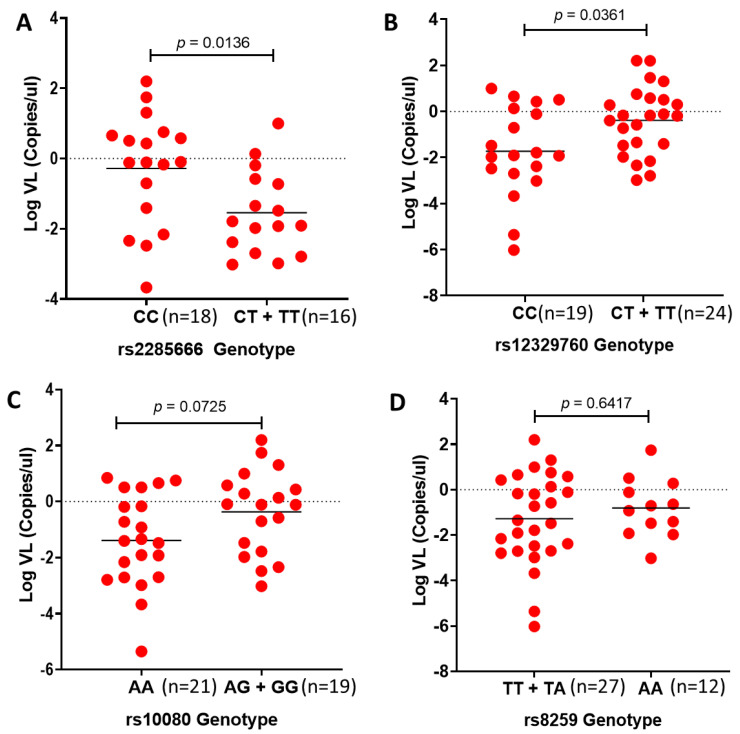
Comparing SARS-CoV-2 viral loads versus rs2285666, rs12327960, rs10080 and rs8259 genotypes in African individuals: Viral loads are correlated with (**A**) rs2285666 and (**B**) rs12329760 but not (**C**) rs10080 and (**D**) rs8259 in African individuals. Mann–Whitney test *p*-value of comparison between homozygous and alternative genotypes.

**Figure 4 genes-14-01798-f004:**
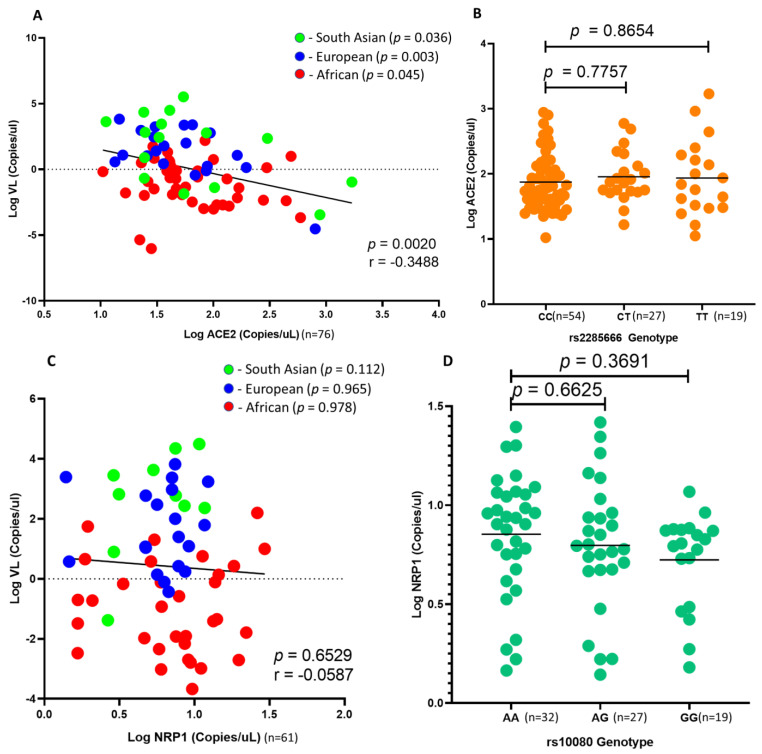
Comparing SARS-CoV-2 viral load with *ACE2* and *NRP1* gene expression. A comparison between *ACE2* and *NRP1* gene expression against rs2285666 and rs10080 genotypes in South African individuals: (**A**) There is a significant linear correlation of SARS-CoV-2 viral load with *ACE2* expression across African (red dots), European (blue dots) and South Asian (green dots) individuals (*p* = 0.045, *p* = 0.003 and *p* = 0.036). However, *ACE2* expression does not correlate with rs2285666 genotypes (**B**). SARS-CoV-2 viral load does not correlate with NRP1 expression (**C**); in addition, NRP1 expression does not associate with rs10080 genotypes (**D**) in South African individuals.

**Table 1 genes-14-01798-t001:** Stratification criteria used to score patients into no clinical presentations and clinical presentations groups.

	No Clinical Presentations (*n* = 133)	Clinical Presentations Present (*n* = 427)
Primary Factors		
Asymptomatic	Yes	No
Symptomatic	No	Yes
Secondary Factors		
Oxygen required	No	Yes
Hospitalised	No	Yes
Pneumonia	No	Yes
Confusion	No	Yes
Sore throat	No	Yes
Loss of taste/smell	No	Yes
Nausea/Vomiting	No	Yes
Headache	No	Yes
Shortness of breath	No	Yes
Body aches	No	Yes
Fatigue	No	Yes
Cough	No	Yes
Fever and chills	No	Yes

Note: Participants were considered to have no clinical presentations if they were asymptomatic, did not require oxygen or hospitalization and did not present with pneumonia. Participants who had clinical presentations were symptomatic and presented with at least one of the secondary factors.

**Table 2 genes-14-01798-t002:** The distribution of African, South Asian and European individuals with no clinical presentations and those with clinical presentations.

Ethnicity	No Clinical Presentations (*n* = 133)	Have Clinical Presentations (*n* = 427)	Total
African	102 (35%)	188 (65%)	290
South Asian	26 (11%)	220 (89%)	246
European	5 (21%)	19 (79%)	24

**Table 3 genes-14-01798-t003:** SNPs within ACE2, TMPRSS2, NRP1 and CD147 used in this study.

Chr	Gene	Locus	rs Number	Variant	Variant Type	MAF (1000 Genome)	Freq in SAP Cohort
x	ACE2	NG_012575.2	rs2285666	C > T	Intron variant	EUR:C = 0.76; T = 0.24 AFR:C = 0.79; T = 0.21 SAS:C = 0.51; T = 0.49	EUR:C = 0.69; T = 0.31 AFR:C = 0.73; T = 0.27 SAS:C = 0.54; T = 0.46
21	TMPRSS2	NG_047085.2	rs12329760	C > T	Missense variant (Valine to Methionine)	EUR:C = 0.76; T = 0.24 AFR:C = 0.71; T = 0.29 SAS:C = 0.77; T = 0.23	EUR:C = 0.77; T = 0.23 AFR:C = 0.78; T = 0.22 SAS:C = 0.78; T = 0.22
10	NRP1	NG_030328.1	rs10080	A > G	3′UTR variant	EUR:G = 0.57; A = 0.43 AFR:G = 0.30; A = 0.70 SAS:G = 0.50; A = 0.50	EUR:G = 0.61; A = 0.39 AFR:G = 0.28; A = 0.72 SAS:G = 0.63; A = 0.37
19	CD147	NG_007468.1	rs8259	T > A	3′UTR variant	EUR:T = 0.70; A = 0.30 AFR:T = 0.44; A = 0.56 SAS:T = 0.37; A = 0.63	EUR:T = 0.50; A = 0.50 AFR:T = 0.40; A = 0.60 SAS:T = 0.19; A = 0.81

Notes: EUR = European, AFR = African and SAS = South Asian.

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
