# Peer review of "Polymorphisms within the SARS-CoV-2 Human Receptor Genes Associate with Variable Disease Outcomes across Ethnicities"

_genes, 2023, doi:10.3390/genes14091798_

Round 1

Reviewer 1 Report

The authors submitted an interesting article for review, however, before its publication it is required to be corrected:

- too briefly described methodology for testing genotypes, expressions or viral load, please specify the descriptions

- the description of the study group is incomprehensible, please correct this part

- in Table 1 - no number of patients in individual subgroups indicated in the Table

- Table 2 - the title of the Table is not fully understood

Reviewer 2 Report

In the present manuscript, the authors studied the genetic variants within the SARS-CoV-2 receptors ACE2, TMPRSS2, CD147 and NRP1. They found there is a significant lower viral load of SARS-CoV-2 in Africans compared to Europeans and south Asians might because the specific SARS-CoV-2 receptor variants within different ethnics.

This manuscript is well written and the data support their conclusion.  
